# CRAgents for Disease Modelling via Multi-Agent Critique and Refinement

**Kelly Yu**[1]                                                     kelly.yu924@gmail.com
**Zuoou Li**[1]                                                        lizuoou@126.com
**Paul Matthews**[2]                                          p.matthews@imperial.ac.uk
**Wenjia Bai**[2]                                                   w.bai@imperial.ac.uk
**Bernhard Kainz**[2,3]                                          b.kainz@imperial.ac.uk
**Mengyun Qiao**[1]                                                m.qiao@ucl.ac.uk

[1] *University College London, London, UK*

[2] *Imperial College London, London, UK*

[3] *Friedrich-Alexander-Universität Erlangen-Nürnberg, Germany*

## Abstract

Automating disease modelling requires systems that can formulate clinically valid prediction tasks and optimise robust models under low-prevalence settings. However, current multi-agent workflows do not unify these stages, leading to clinically weak target selection and reliance on manual model configuration. We propose **CRAgents**, a multi-agent framework that couples task formulation with model optimisation within a unified workflow. Built from eight agents, it introduces two reasoning loops: (i) a debate-driven loop (Inner Debate Loop) that performs *critique* through systematic, clinically grounded evaluation of modelling targets; and (ii) a reflection-driven loop (Outer Reflection Loop) that enables *refinement* through iterative model improvement based on validation behaviour. We evaluate **CRAgents** on the UK Biobank dataset, where it outperforms recent agent-based baselines, achieving an AUC of 0.781, demonstrating that the combination of critique and refinement improves reliability and clinical alignment in automated disease modelling.

**Keywords:** multi-agent systems, automated machine learning, LLM agents

## 1. Introduction

Building a clinical prediction model involves several decisions before any model is trained: which outcome to predict, which variables to use as predictors, and how to evaluate performance in a way that matches clinical use (Collins et al., 2024; Efthimiou et al., 2024). These decisions get harder when cohorts are large and features are high-dimensional, as is now common with imaging-derived data (Aerts et al., 2014), and audits of published ML prediction models have repeatedly flagged poor methodological choices in this space (Wolff et al., 2019). Automating the pipeline alone is insufficient, as the validity of upstream decisions must also be maintained under automation (Gorenshtein et al., 2025). The challenge is most pronounced for rare outcomes (Salmi et al., 2024). A target may be statistically learnable yet clinically uninformative, and as prevalence decreases, performance becomes highly sensitive to thresholding, resampling, and validation choices (Van den Goorbergh et al., 2022). Existing multi-agent systems (MAS) automate parts of the pipeline (Zhao et al., 2026; Feng et al., 2025), but typically treat the decisions of *what* to predict and *how* to train as separate problems (Chang et al., 2024). As a result, target selection often remains weak and model configuration still requires human intervention. We propose CRAgents, a multi-agent framework that unifies these decisions into a single optimisation problem.

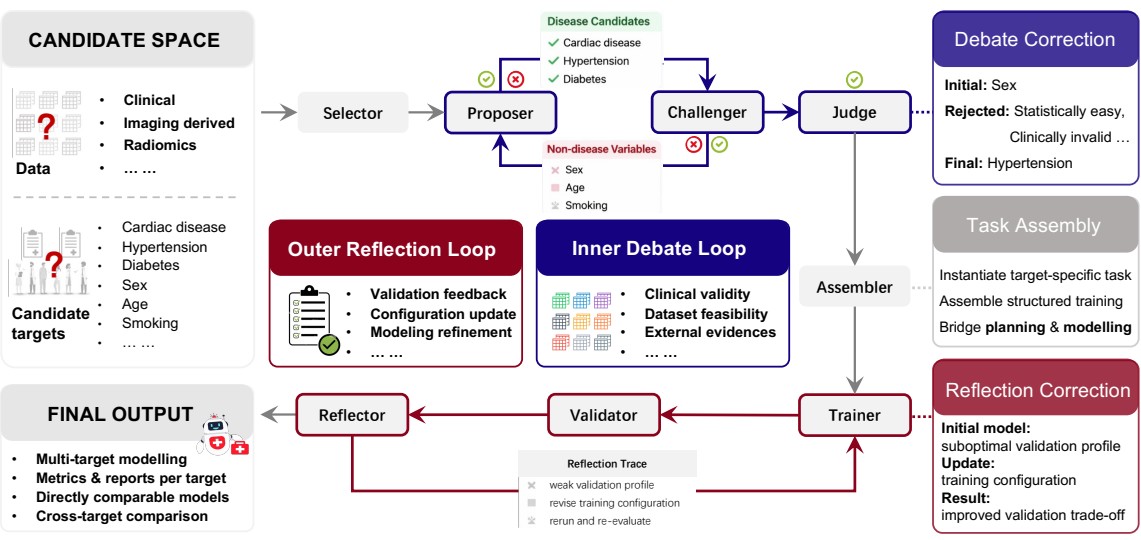

Figure 1: CRAgents framework for automated disease modelling through multi-agent reasoning. The workflow includes eight agents, with the Inner Debate Loop and Outer Reflection Loop serving as two key reasoning processes.

## 2. Methods and Results

Let $D = \{x_i\}_{i=1}^n$ be a cohort dataset with patient features, and let $\mathcal{Y}$ denote the set of candidate endpoints available in the cohort (e.g., clinical outcomes, imaging-derived labels, and non-disease variables). Given a user requirement $r$ and a clinically admissible subset $T \subseteq \mathcal{Y}$, we aim to (i) select a disease-relevant endpoint $t^\star \in T$, optionally with a ranked shortlist $\mathcal{P}^\star \subseteq T$ for multi-disease modelling, and (ii) train, for each selected target $t$, a model $f_{\theta_t^\star}^{(t)}$ under a unified evaluation protocol, so that performance can be compared across targets.

**Dataset** We evaluate CRAgents on a subset of 26,894 UK Biobank (Sudlow et al., 2015) participants with demographic, lifestyle, and imaging-derived clinical variables, covering disease endpoints such as cardiac disease, hypertension, and diabetes.

**CRAgents workflow** Agentic pipelines for disease modelling tend to fail in two ways: they do not adequately validate target choice and they iterate on model configurations without a clear objective. CRAgents addresses both through a two-stage pipeline (Fig. 1) built from eight agents: *Selector*, *Proposer*, *Challenger*, *Judge*, *Assembler*, *Trainer*, *Validator*, and *Reflector*. A debate-driven loop first determines the target: the Selector enumerates candidate endpoints, the Proposer and Challenger debate admissible targets from opposing angles, and the Judge selects a primary endpoint $t^\star$ and a ranked shortlist $\mathcal{P}^\star$. A reflection-driven loop then refines the model for each target: the Assembler constructs a model-ready task with fixed label definitions and splits; the Trainer fits models, the Validator performs stratified cross-validation, and the Reflector proposes updated configurations based on it-

Table 1: **Quantitative comparison on Heart (UKBB) (Sudlow et al., 2015).** Best in **bold**; second-best underlined.

|  | Method | AUC ↑ | Recall ↑ | FNR ↓ |
|---|---|---|---|---|
| *Baselines* | M$^3$Builder (Feng et al., 2025) | 0.744 | 0.528 | 0.472 |
|  | MESHAgents (Zhang et al., 2025) | 0.752 | 0.556 | 0.444 |
| *Ablations* | + Debate Loop | 0.773 | 0.541 | 0.459 |
|  | + Reflection Loop | 0.729 | 0.587 | 0.413 |
|  | **CRAgents (Full)** | **0.781** | **0.610** | **0.390** |

eration history. The pipeline returns models and validation reports that are comparable across all endpoints in $\mathcal{P}^\star$.

**Experiments and Results** All methods use the same preprocessing, training backend, data split policy, and validation procedure. Model performance is evaluated using AUC, Recall, and false negative rate (FNR). We evaluate CRAgents against two recent multi-agent baselines, M$^3$Builder (Feng et al., 2025) and MESHAgents (Zhang et al., 2025), adapted to the same target space and training budget without manual reconfiguration, and run two ablations (+Debate Loop, +Reflection Loop) to isolate each reasoning loop.

Table 1 shows CRAgents outperforms both baselines on every metric, improving AUC from 0.752 (MESHAgents) to 0.781, cutting FNR from 0.444 to 0.390, corresponding to an approximately 12% reduction in missed positives. We evaluate robustness across five backends (GPT-4o mini, GPT-4.1 nano, Claude Haiku 4.5, Claude Sonnet 4.5, and DeepSeek V3.2), obtaining a mean AUC of $0.782 \pm 0.012$, which indicates stable performance across model choices.

## 3. Conclusion, Limitations and Future Work

We presented CRAgents, a multi-agent framework for automated disease modelling that couples task formulation with model optimisation through two reasoning loops: a debate-driven loop for clinically constrained target selection and a reflection-driven loop for validation-guided optimisation under low-prevalence settings. On UK Biobank, these loops deliver complementary gains by improving discriminative validity and reducing missed positives. Our modular design enables explicit reasoning over both stages but introduces a stage-wise decomposition of the workflow. As a result, coordination between planning and modelling remains structured rather than fully joint, which may limit end-to-end optimisation. Future work will explore more integrated coordination strategies that preserve transparency while enabling richer cross-stage optimisation, as well as extending the framework to higher-dimensional radiomics data and sequential multi-target modelling.

## Acknowledgments

This work used data from the UK Biobank under Application Number 18545. We thank the UK Biobank participants and staff.

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
