# OpenReview forum: "CRAgents for Disease Modelling via Multi-Agent Critique and Refinement"
_MIDL.io/2026/Short_Papers — MIDL 2026 - Short Papers Poster_

### Official Review · Reviewer_JdRs · 2026-05-06
**A multi-agent framework for jointly determining best target/model for disease modeling**

**Rating:** 4
**Confidence:** 4

**Review:**

Quality: Motivation for the proposed framework makes sense (determine best prediction target and model in one framework), experiments include comparison to 2 recent baselines and using a very large public dataset.

Clarity: Paper writing is mostly clear, though a few points could use clarifying (see below).

Originality: Many multi-agent frameworks exist, but I think this is new.

Significance: The proposed method provides an automated solution toward the problem of how to define model targets and find the best models for a specific disease of interest.

**Summary:**

This paper presents CRAgents, a multi-agent framework for disease modeling that determines the best target for model prediction along with training of the best model in a single workflow. It proposes two reasoning loops, 1) a debate loop that decides on the modeling target, and 2) a reflection loop that performs the model optimization for the selected targets. The approach is tested on UK Biobank data for heart disease modeling and compared favorably to two recent multi-agent frameworks.

**Strengths:**

1. The paper does motivates well the need of the proposed framework, and makes the 2 part reasoning structure makes sense.

2. The experiments are performed on the large UK Biobank dataset, and comparisons against 2 recent baselines are presented, with performance measured using 3 different metrics. The proposed method appears to outperform the 2 baselines by a good margin.

3. Ablation studies were performed to assess the framework design of the 2 reasoning loops.

4. The paper writing and flow are clear. Fig. 1 is very informative for understanding the proposed approach.

**Weaknesses:**

1. Some additional details about the experimental settings would be useful - how much data is used for training/testing?

2. Would be nice to see some additional analysis of the results, e.g., what targets did the debate loop end up choosing, as short list candidate and primary target? Did these make "sense"?

3. What exactly does a no debate loop setting represent - does this mean that every target possible is used for modeling? If this is the case, wouldn't this end up finding the best performing model, since you tested models for all possible targets? I might be misunderstanding.

4. Some additional, earlier definition of the disease being modeled would be helpful - I don't think I understood that the prediction had to do with heart until I referred to table 1. Furthermore, not sure exactly what the FNR / missed positives and recall performance metrics are in reference to, since we are not told the exact target that was used for modeling, which makes it hard to assess the significance of these values.

**Justification Of Rating:**

This paper presents timely work on using multi-agent framework in disease modeling and prediction. Clarification of some missing/confusing details would further improve the ability for readers to understand the work and results.

---

### Decision · Program_Chairs · 2026-05-08

Accept (Poster)